# Effects of Linkers and Substitutions on Multitarget Directed Ligands for Alzheimer’s Diseases: Emerging Paradigms and Strategies

**DOI:** 10.3390/ijms23116085

**Published:** 2022-05-29

**Authors:** Narayanaperumal Pravin, Krzysztof Jozwiak

**Affiliations:** Department of Biopharmacy, Medical University of Lublin, Chodzki 4a, 20-093 Lublin, Poland

**Keywords:** Alzheimer’s disease, multitarget directed ligands, cholinesterase, amyloid-β-fibrils, monoamine oxidase B

## Abstract

Alzheimer’s disease (AD) is multifactorial, progressive and the most predominant cause of cognitive impairment and dementia worldwide. The current “one-drug, one-target” approach provides only symptomatic relief to the condition but is unable to cure the disease completely. The conventional single-target therapeutic approach might not always induce the desired effect due to the multifactorial nature of AD. Hence, multitarget strategies have been proposed to simultaneously knock out multiple targets involved in the development of AD. Herein, we provide an overview of the various strategies, followed by the multitarget-directed ligand (MTDL) development, rationale designs and efficient examples. Furthermore, the effects of the linkers and substitutional functional groups on MTDLs against various targets of AD and their modes of action are also discussed.

## 1. Introduction

For a long time, the paradigm of “one-drug, one-target, one disease” has had countless successes for the pharmaceutical industry in the 20th century [1]. Many promising drugs have been acknowledged from this framework, which will predominate certain diseases in the future [2]. In contrast, this framework is not effective enough to cure multifactorial diseases such as neurodegenerative diseases [3], cancer [4] and infections [5], due to the fact that their onset and progression involve multiple proteins, receptors or signaling pathways. Alzheimer’s disease (AD) is one of the progressive multifactorial neurodegenerative disorders that has developed serious global health concerns amongst the elderly [6]. This lethal disease is the main cause of dementia, which includes the clinical symptoms of progressive memory loss, impaired executive functions and difficulties in performing routine daily activities.

In 2021, the World Health Organization (WHO) data projected that 55 million people were affected with dementia worldwide, and this figure is expected to escalate to 78 million in 2030 and 139 million in 2050 [7]. The precise physiological factors of this multifactorial disease are still not fully understood. Researchers have developed diverse hypotheses and shed light on several factors of AD onset and progression. One of those hypotheses is cholinergic dysfunction, according to which, the gradual degradation of cholinergic neurons and their functions occur; thus, cholinergic signaling and neurotransmission in the brain may be affected [8]. Alternatively, the amyloid cascade hypothesis describes that amyloid fibrils caused by β-amyloid proteins in the brain are responsible for the onset of AD, which is the result of abnormal processing of the amyloid precursor protein (APP) by β- and γ-secretase enzymes. These amyloid fibrils interfere with neuronal functions and lead to brain degeneration [9]. Another hypothesis assumes an induction of excessive oxidative stress. The excessive accumulation of metals such as copper, zinc, iron, aluminum and mercury is accompanied by free radicals or reactive oxygen species (ROS). Their generation in the brain is actively involved in lipid peroxidation and protein oxidation, and the stress developed by these events opens up the way for AD onset [10]. Currently, few drugs (Figure 1) are available on the market to treat AD; however, they are unable to cure it completely, and only symptomatic treatments exist.

To overcome the drawbacks of the existing drugs, a “one-drug, multi-target” paradigm has been developed for AD candidates. Increasing amounts of evidence indicate that the multi-target approach may improve therapeutic efficacy and safety [11,12,13,14], and only a few of them are on the market or in clinical trials, despite the best efforts of medicinal chemists [15,16]. The major molecular targets of AD for MTDLs are depicted in Figure 2. In recent years, an enormous number of articles have been published describing, specifically, developed compounds exhibiting multitarget profiles [13,14,17,18], suggesting the multitarget-directed ligand (MTDL) strategy as fruitful and worthy of further exploration. In this review, the effects of choosing linkers and functional groups on the multitarget-directed ligands for effective AD targets are summarized. The synthetic strategies of the development of MTDLs and the mechanisms of their actions on various AD targets are discussed as well.

### 1.1. The Cholinergic Hypothesis

The cholinergic hypothesis concerns major neurotransmitters and cholinesterases (acetylcholinesterase (AChE) and butyrylcholinesterase (BuChE)), which are extracellular enzymes present abundantly in the central nervous system and at peripheral junctions. The reactions between the choline acetyltransferase (ChAT) enzyme and acetyl-coenzyme A (Ac-CoA) yield acetylcholine (ACh). The nicotinic receptors and muscarinic receptors play a key role in cholinergic transmission and, among other important functions, are responsible for learning and memory generation in the brain. Thus, the maintenance of ACh levels in the brain is essential for proper brain function. AChE is a serine hydrolase enzyme that decomposes neurotransmitter ACh at the neuronal synapses into acetic acid and choline. The choline is recycled and reused by the presynaptic neuron. In AD patients, the brain-elevated activity of AChE is observed, which reduces the required Ach levels in the brain, impairs synaptic functions and eventually produces toxicity in neuronal cells, leading to brain damage, as well as memory loss and dementia [19]. It is worth highlighting that AChE is the main therapeutic target within the cholinergic theory.

### 1.2. The Amyloid Cascade Hypothesis

Amyloid-β-peptide (Aβ) is the connective agent in AD pathology. Neurofibril tangle formations, vascular damage and dementia follow as a result of Aβ deposition in the brain. This was postulated by Hardy and Higgins in 1992. This hypothesis has dominated the understanding of AD for more than two decades. It states that the pathogenic procedure begins with abnormal processing of the amyloid precursor protein (APP) [20]. The proteolytic cleavage of APP by the β-secretase (BACE-1) complex produces 99 amino acid-long APP carboxy-terminal fragments that can be further cleaved by γ-secretase at the transmembrane segment. That last step can generate Aβ peptides with various chain lengths, including Aβ37, 38, 39, 40, 42 and 43 [21,22]. Among them, Aβ42 and Aβ40 comprise the two major Aβ species in the brain. Compared to soluble Aβ40, Aβ42 has a higher tendency toward aggregation, due to the hydrophobicity within its two terminal residues. Initially, Aβ42 start to aggregate into oligomers that can interfere with cholinergic neurotransmission in the brain. Both Aβ and tau amyloid aggregates are manifestations of the improper function of the neurons, which leads to AD symptoms [23]. Thus, hyperphosphorylated tau proteins further generate neurofibrillary tangles that interfere with intracellular metabolism, eventually leading to neuronal death. Recently, Aducanumab (anti-Aβ monoclonal antibody) was approved for medical use in the United States by the FDA in June 2021 [24].

### 1.3. The Oxidative Stress Hypothesis

Reactive oxygen species (ROS) are the main cause of oxidative stress in cells. Generally, ROS are highly reactive and unstable chemical species because they have one or more unpaired valence shell electron. These species are derived from oxygen molecules such as the hydroxyl radical (^•^OH) or the superoxide anion ($O_{2}^{-}$). ROS generation is a normal cellular function in a living organism, and it is an outcome of the cellular metabolism that is substantial to the maintenance of hemostasis. The concentration level of ROS production is very important for various physiological functions, and abnormal ROS production or oxidative stress plays a key role in AD development [25]. These ROS can be generated in the brain via exogenous sources (exposure to environmental factors, pharmaceuticals, etc.) or endogenous sources (mitochondrial and non-mitochondrial enzymes, including nicotinamide adenine dinucleotide phosphate—NADPH oxidase (NOX), xanthine oxidase (XO), granular endoplasmic reticulum cytochrome P450 and flavo-oxidase). The main source of endogenous generation is the respiratory chain and reduction-oxidation systems [26]. The superoxide radical, for example, is the most common; its main source of generation is the electron transport chain [27]. However, the unstable species become more stable themselves after oxidizing biomolecules that can later serve as biomarkers of oxidative damage (e.g., lipids, nucleic acids and proteins in their oxidized states). Altered levels of antioxidant enzymes, such as catalases, superoxide dismutase and glutathione peroxidase, as well as other antioxidant substances, can also be considered as biomarkers [28]. For these facts, antioxidant species are promising agents in the prevention and treatment of oxidative stress-related diseases. It is important to mention that antioxidants are species that can sequester free radicals, thus breaking the radical chain reaction and stabilizing both molecules [29].

Moreover, the disturbed metabolism of redox-active (Fe and Cu) and redox-inactive (Zn) metals are involved in oxidative stress. Normally, the bio-essential metals Fe, Cu and Zn are present in the neocortex and play several critical roles in the brain. They are cofactors for numerous enzymes with important catalytic activities (e.g., energy production) and serve as active participants in metal-dependent neurotransmission. Redox-active metals act as catalysts to generate free radicals in addition to the Fenton reaction. The generated free radicals increase the oxidative stress of the brain cells. In addition to that, redox-active metals promote amyloid fibril formation in the brain by the interaction with the amino acid residues. A therapeutic approach would consist of the use of small molecules (metal chelators) to deplete the excess Cu, Zn and Fe [30]. Other hypotheses explaining AD-related pathologies also exist. They have been discussed in numerous excellent articles [31].

## 2. Design and Development of Multitargeted Directed Ligands (MTDLs)

A recent strategy to achieve better AD drugs is to simultaneously tackle several AD-related pathological features [32]. Therefore, researchers have switched the new avenue of multitarget-directed molecules that possess various AD pathogenesis-targeting chemical entities engineered as a single molecule. The single multitarget-directed molecule has several benefits compared to several one-target agents. Predominantly, multifunctional drug molecules can be characterized by increased therapeutic potential through synergies even at low doses and [33] reduced harmful drug–drug interactions [34]. Most importantly, the administration of MTDLs is more convenient, as it requires a single medication, which is a significant advantage for AD candidates affected by memory loss, since it facilitates patient compliance [35]. To make the MTDLs, various strategies have been developed and successfully synthesized for AD treatment [36,37,38,39].

### 2.1. Rational Design

The basic rational design of MTDLs is to take two or more chemical entities with biologically known targets (e.g., cholinesterase, monoamine oxidases, Aβ fibrils and metal ions) and combine them into a single chemical entity [40]. In the number of chemical entities combinations, the MTDLs are classified into linked, fused and merged types (Figure 3). In the linker approach, the pharmacophores are separated by a linker that does not exist in the parent molecules [41]. To ensure the original scaffold activity, the length, position and composition of the linkers should be optimized properly. The fusion strategy is an extension of the linker approach, which means that the linker length is zero and that the two pharmacophores are directly connected. As compared to linked pharmacophores, the molecular weight of the MTDLs is slightly lower due to the removal of linkers and, hence, may favor BBB permeability. However, this imparts less flexibility to each moiety to move into its target binding pocket. The most beneficial approach is merging, which involves two or more different targeted pharmacophores overlapped fully; therefore, the molecular size or weight is lower than the linked and fused pharmacophores, which leads to attractive pharmacokinetic properties such as solubility, bioavailability and BBB permeability. These are the favorable MTDLs in terms of drug-likeness, but are the least likely to retain function; however, that may be addressed by multiple rounds of design [42].

### 2.2. Linked MTDLs for AD Targets

The linking strategy is most common for constructing MTDLs against AD. The selection of linkers, substituted functional groups, position of tailoring on the two distinct pharmacophores (should not join a ligand at the position with a steric effect) and length of the linkers should maintain the activity of the MTDLs. Most importantly, linkers themselves may sometimes interact with the target residues [43]. Overall, the linkers are the backbone of this strategy; hence, the selection/design of the linkers is very important to achieve better-linked MTDLs. In this part, a few significant linked MTDLs are reviewed. Unlike other commercially available inhibitor drugs, galantamine (**1**) has a postulated dual mode of action: in addition to inhibiting AChE, it has a facilitating effect on the nicotinic receptor-mediated transmission via allosteric modulation of the α-subunit of nicotinic receptors. In particular, **1** can enhance synaptic N-methyl-D-aspartate receptor (NMDAR) activity by activating nicotinic receptors located on presynaptic glutamatergic neurons. On the other hand, Memantine (**2**) is the only clinically available NMDAR antagonist and is prescribed to treat moderate-to-severe dementia. Moreover, **1** has a memory-enhancing activity, and it can also utilize the complement of the effect of **2** in AD [44]. Based on these perspectives, Simoni et al. developed for the first time a series of MTDLs with the combination of these two marketed drugs (**1** and **2**) linked by flexible spacers (Figure 4A). Interestingly, one of the synthesized compounds, i.e., **3**, showed nanomolar inhibition for AChE (IC_50_ = 1.16 nM), micromolar-binding affinity for NMDAR (K_i_ = 4.6 µM) and NMDAR-containing 2B subunit (NR2B) (ifenprodil-binding assay, K_i_ = 4.6 μM). It was observed that the five to six methylenes units were the minimal distance to contact both active sites (the internal and the peripheral anionic sites) of AChE; when the spacer length of **3** was reduced to four carbons, the inhibition efficacy decreased significantly (IC_50_ = 695.9 nM). Moreover, the linker length did not affect the binding affinity against NMDAR. In addition to that, **3** was able to inhibit NMDA-induced neurotoxicity with sub-nanomolar potency and exhibited remarkable neuroprotective activity (cell-based assay, IC_50_ = 0.28 nM) compared to marketed drugs **1** (IC_50_ = 747 nM) and **2** (IC_50_ = 718 nM) [45].

Eugenie et al. synthesized MTDLs based on the standard drug tacrine (**4**) in combination with Trolox (**5**) by coupling reactions for various AD targets (Figure 4B). Compound **6** showed an improved inhibitory profile in vitro against AChE and moderate antioxidant activity. Moreover, the modifications of the linker length did not much affect the inhibition activity of hAChE. According to the IC_50_ value, the unit of three or six carbon (IC_50_= 80 nM) was the optimum length for hAChE, and the activity was even 125-fold better than that of **4** and **5**. On the other hand, the unit of four carbon-length linkers showed slightly more activity against hBuChE. Further, the linkers themselves interacted hydrophobically with the surrounding hydrophobic amino acid residues (e.g., Phe297, Tyr341 and Tyr124), which was identified by molecular modeling. The flexibility and length of the linker played a major role in making compound **6** to reach both the catalytic site and peripheral anionic site of hAChE. In addition to that, the antioxidant activity of compound **6** (DPPH, IC_50_ = 44.09 µM) was reported to be due to the structural feature of the Trolox moiety, as **5** is a standard antioxidant. The authors suggested that compound **6** could be a suitable candidate for mild to severe forms of AD [46].

### 2.3. Fused MTDLs for AD Targets

In the fusion strategy, two distinct pharmacophores fused chemically using various reaction conditions. The active functional groups should be maintained to retain the activity against various targets, and the developed MTDLs are smaller than the linked pharmacophores and provide better access to the pharmacokinetic profile. By adopting this strategy, compound **9** was developed by fusing the benzofuran ring from Moracin M (**7**) (Phosphodiesterase 4D (PDE4D) and Aβ inhibitor) and the hydroxyquinoline ring from clioquinol (**8**) using Witting regents. The synthesized compound **9** exhibited significant inhibitory potency towards PDE4D (IC_50_ = 0.32 μM), antioxidant activity and self- and metal-induced Aβ aggregation properties (Figure 5A). Notably, the biological potential was reduced, while the phenolic hydroxyl group was removed from the benzofuran moiety. Moreover, the substitution position of the hydroxy group on benzofuran (R^3^) played a vital role in its biological activity, particularly Aβ aggregation inhibition (67.5%), which was much higher than Moracin M (43.1%) [47].

A series of fused MTDLs with the combination of clioquinol (**10**) and donepezil was (**11**) [48] prepared and structurally characterized by Federica et al. (Figure 5B). The authors followed a systematic synthetic protocol that started with the reaction of piperazine and 8-hydroxyquinoline and achieved the intermediate 7-(piperazin-1-ylmethyl)-8HQs via a multicomponent Mannich reaction. To achieve a good yield and reduce the reaction time, they utilized microwave-assisted conditions (130 °C, 45 min) rather than the conventional heating-based procedures. In the end, the target compounds were synthesized through classical S_N_2 nucleophilic substitution with excellent yield (47–97%). The most active compound, **12**, was identified by the initial biological characterization, and it is worth noting that all the fused pharmacophore series were shown to be weak inhibitors of hAChE, which did not exceed 21% in the 100-µM range; in the case of hBChE, all the compounds showed a significant inhibitory profile (9–89.1%, in a 40-µM range). The substituted functional groups (chloro, methoxy, etc.) and the position of substitution (2, 3 or 4) on the benzyl moiety significantly influenced the inhibition profile of hBChE. For example, when the position of the methoxy group moved from position 2 to 3 or 4, the inhibitory efficacy (IC_50_ = 47.2 µM) was eight times lower than the substitution of position 2 (**12a**, IC_50_ = 5.71 µM). On the other hand, the substitutional functional group and position of the substitution did not influence the Aβ42 self-aggregate inhibition (38.1–59.4%). Compared to **12a**, compound **12b** showed potent inhibitory activity for the Aβ42 amyloid (65%), and the value was closer to the known antiaggregating agent curcumin (73.7%). Other serious MTDLs were synthesized via the fusing strategy (Figure 5C), and the metal chelating agent clioquinol and the antioxidant ebselen were fused by a sequence chemical reaction and obtained MTDLs with excellent yield (75–96%). Compared to their parent compounds, clioquinol and ebselen, these synthesized MTDLs exhibited significant potency in inhibiting the aggregation of self- and Cu(II)-induced amyloid-β (Aβ) and acted as remarkable antioxidants. Interestingly, compound **13** was found to possess rapid H_2_O_2_ scavenging activity and glutathione peroxidase-like (GPx-like) activity. The type of substitution group at the oxine moiety played a major role in GPx-like activity, e.g., the chlorine substituted showed much better GPx-like activity as compared with the iodine substitution. Along with this, **13** could be able to form complexes with biometals such as Cu, Zn and Fe with the stoichiometry of a 1:2 Cu^2+^:ligand molar ratio. According to the results of the in vitro blood–brain barrier model assay, **13** was able to penetrate the CNS, and the toxicity tests in mice showed that **13** had no acute toxicity at doses of up to 2000 mg kg^−1^ [49].

### 2.4. Merged MTDLs for AD Targets

Due to the low molecular weight and efficient pharmacokinetic profile, the merged MTDLs have received much attention in AD drug design. A series of 3,4-dihydropyrimidin-2(1H) derivatives (calcium channel blocker) and tacrine (ChEs inhibitor) were merged via the Friedlander-type reaction by Bartolini et al. They investigated the detailed ChE inhibition and calcium intake blockade profile. All the compounds showed moderate-to-good ChEs inhibition, and the most important compound was **14**, which showed balanced ChE inhibition (IC_50_ = 3.05 µM and 3.19 µM for AhChE and hBuChE, respectively, and calcium channel blockade activity (30%) (Figure 6A). It was interesting to observe that **14** did not show any hepatotoxicity towards HepG cells, such as tacrine. Indeed, the predicted BBB permeability and pharmacokinetic profile provided the value of the compound for further modification or biological screening [50]. Sheng et al. synthesized a series of novel 1-phenyl-3-hydroxy-4-pyridinone derivatives (Figure 6B) from the inspiration of the aminopropyl phenyl moiety (H_3_ receptor antagonism), SKF-64346 (Aβ aggregation inhibitor) and deferiprone (metal chelation and radical scavenger) by using the merging strategy. The synthetic conditions involved multiple steps; the initial intermediates were achieved by a microwave apparatus with 155 °C heating for around 10–20 min. The final compounds were synthesized with a good yield (93–99%) in the presence of the pd/C catalyst. Most of the compounds exhibited antagonism toward the H3 receptor, Aβ aggregation inhibition, metal ion chelation and radical scavenging activity. Especially, the most promising compound, **15**, displayed nanomolar IC_50_ values in H3 receptor antagonism (IC_50_ = 0.32 nM) with high selectivity, efficient capability to interrupt the formation of Aβ_1–42_ fibrils (IC_50_ = 2.8 µM), good copper and iron-chelating properties and more potent 2,2′-azinobis(3-ethyl-benzothiazoline-6-sulfonic acid) radical cation (ABTS^•+^) scavenging activity. Significantly, **15** did not show any significant toxicity in human glioma U251 cells at a concentration of 20 µM. In vivo data provided the BBB penetration ability of **15**. Overall, **15** was able to hit four important AD targets [51].

## 3. Examples of MTD Pharmacophores

Numerous MTDLs for AD have been reported in the literature, and a few efficient MTDLs are discussed in this part (Figure 7). Sun et al. synthesized a series of tacrine-based MTDLs to hit two crucial AD targets, such as glycogen synthase kinase-3β (GSK-3β) and human acetylcholinesterase (hAChE). The synthetic steps involved acylation, addition, cyclization, Mitsunobu reaction and amination. The authors analyzed GSK-3β enzymatic inhibition activity of the MTDLs on human recombinant GSK-3β by using the luminescence method. One of the synthesized compounds, **16**, showed a nanomolar inhibition (IC_50_ = 66 nM) profile on GSK-3β, and the activity was improved significantly while increasing the length of the linker from two to five carbon units. Ellman’s test showed that **16** inhibited hAChE in the nanomolar range (6.4 nM). It was interesting to note that the substitution on the tacrine moiety (6-Cl-tacrine or 7-MeO-tacrine) increases the inhibitory activity of hAChE more compared to BuChE, suggesting that such a modification could help to design highly selective hAChE inhibitors. The binding mechanism of **16** with hAChE was identified by molecular docking data, which revealed that **16** occupied both the catalytic active site (CAS) and peripheral anionic site (PAS). Endocyclic nitrogen was involved in the hydrogen bond interaction with the mainchain carbonyl oxygen of His447 (catalytic residue); thus, the catalytic triad was disrupted. The authors found that the pyridothiazole moiety was located in the PAS of the hAChE-binding groove and formed π–π stacking contacts with the side chain of Trp286. In addition to that, the cognition-improving potency of **16** was assessed by the Morris water maze test, and the results indicated that **16** showed better cognitive improvement and less hepatotoxicity than tacrine [52]. Few tacrine-hydroxamate-based MTDLs were synthesized, and their efficacy was investigated on the AD targets such as ChEs, histone deacetylases (HDACs), Aβ1-42 fibrils and metal ions. Compound **17** was synthesized by reacting intermediate (E)-ethyl3-(4-((3-((6-chloro-1,2,3,4-tetrahydroacridin-9-yl)amino)propyl)carbamoyl)phenyl)acrylate with NH_2_OH in the presence of KOH in MeOH. All the tested compounds showed better ChEs inhibitory activity than the standard (tacrine) in the nanomolar range. Compound **17** was found to show the most potent inhibition for AChE (IC_50_ = 0.12 nM); along with the superior selectivity of ChEs, the tacrine unit could be responsible for the inhibition of ChEs. When the spacer between N and hydroxamate is longer and the substitution of chlorine at the sixth position in tacrine benefits the inhibition of AChE, the inhibitory activity of BuChE remains comparable. Interestingly, when the connecting amide bond between N and hydroxamate was changed to a sulfonamide bond, the inhibition on AChE was decreased while the inhibition on BuChE was increased, and a similar pattern was observed on HDAC inhibition, such as exchanging the amide (IC_50_ = 0.23 nM) to sulphonamide (IC_50_ = 6810 nM) to reduce the inhibitory activity. Significantly, compound **17** was proposed as a mixed-type inhibitor, since it could bind CAS and PAS, which is evident from the kinetic data and molecular modeling studies [53].

Wieckowska et al. synthesized 12 MTDLs based on indole scaffold and investigated their biological activities against selected AD targets, including BuChE, serotonin HT_6_ antagonism and antioxidant properties [54]. To isolate compound **18**, a systematic synthetic protocol was followed. In brief, the key intermediate of 4-(1-benzyl-1H-indol-4-yl)piperazin-1-hydrogen chloride was alkylated in acetonitrile reflux with ω-bromoalkylphthalimides and K_2_CO_3_ as a base (yield = 73%). From these MTDLs, compound **18** exhibited dual activity against selected targets: 5-HT_6_ receptor (Ki = 41.8 nM) and BuChE (IC_50_ = 5.07 μM). The authors noted that the length of the alkyl linkers affected the affinity of the compounds for the 5-HT_6_ receptors. For example, the elongation and extension of the linkers drastically reduced the affinity, which was reflected in the results of the radioligand-binding assay. The same results were observed for the BuChE inhibitory profile as well; few longer-length linker compounds displayed modest activity or appeared inactive at the screened concentration. The favorable antioxidant property of **18** was screened by a FRAP assay, and **18** exceeded the activity of ascorbic acid at the corresponding concentration; the indole moiety could be the reason for this, as was previously confirmed in numerous studies. The authors suggested that compound **18** might be a promising scaffold for starting further modifications toward the development of MTDLs against AD [55].

## 4. Effects of the Linkers and Substitutions on Pyrazolopyrimidinone Derived MTDLs

Pyrazolopyrimidinone is a nitrogen-containing heterocyclic compound that acts as a building block for several pharmaceutically relevant compounds. Pyrazolopyrimidinones showed excellent biological activity against various diseases, such as cancer, obesity and cystic fibrosis [56]. In addition to that, the derivatives acted as a good inhibitor of phosphodiesterase-9 (PDE9), another promising AD target [57]. Luo et al. generated a series of pyrazolopyrimidinone–rivastigmine hybrids (with and without carbamate moiety) to study the inhibitory activities against PDE9A and ChEs (Figure 8A). This study showed that, compared to the hydroxy substitution (**19a**, IC_50_ = 35nM), the carbamate (**19b**, IC_50_ = 14 nM) substitution at the fourth position of the phenyl ring significantly improved the PDF9A inhibitory activity. A similar trend was followed by the hybrids in ChE inhibition; of note is that **19b** exhibited better selectivity against BuChE (IC_50_ = 3.3 mM) than AChE [58]. The nanomolar inhibition and the negligible cytotoxicity against the SH-SY5Y cell line proved the values of the compounds for further explorations. Analogs of **19** were developed by Huang et al., but in those analogs, pyrazolopyrimidinone was retained and the benzyl piperidine moiety (donepezil) fused via various linkers (Figure 8B). The lengths of the linkers and the substitutions in the linker played a key role in the inhibition activities; the secondary amine (-NH) alkyl linker with two carbon units hybrid **20a** showed better inhibitory activity against AChE (IC_50_ = 0.298 μM) than the one-carbon units linker hybrid (IC_50_ = 0.340 μM). The inhibition activity was slightly increased, while the -N atom was replaced by an O atom (**20b**, IC_50_ = 0.233 μM) and drastically increased with the CH_2_ molecule (**20c**, IC_50_ = 0.048 μM). Of significance, the favorable linker length between piperidine and pyrazolopyrimidinone to inhibit PDE9A was four atoms (**20b**, IC_50_ = 0.285 μM); the activity was decreased when the linker length increased to five or decreased to three atoms. The substitution in the linker showed minor effects on the PDE9A inhibitory activity [59]. Taken collectively, the results indicated the importance of the linker and type of substitution, and most important was the selection of the chemical structures. For example, the rivastigmine scaffolds (**19a**,**b**) did not show any significant inhibition against AChE, while the donepezil scaffold-installed compounds (**20a**–**c**) showed significant inhibition against AChE.

## 5. Effects of the Linkers and Substitutions on Donepezil-Derived MTDLs

Donepezil is a rapidly reversible inhibitor of AChE approved for the treatment of AD, and it is the first and only AChE inhibitor approved for the treatment of severe AD. Donepezil derivatives and hybrids target various AD-related targets, such as BuChE, Aβ cascade and metal ions [31,60,61,62,63,64,65]. Ma et al. generated a series of deoxyvasicinone–donepezil hybrids **21**, in which the piperidine fragment of donepezil was replaced by the piperazine moiety and studied the inhibitory activities against AChE, BACE-1 and the Aβ protein (Figure 9A). The hybrids are selective towards AChE over BuChE, and the IC_50_ values are in the nanomolar range (AChE, **21a** IC_50_ = 3.29 nM; **21b** IC_50_ = 56.14 nM). The results of the study clearly showed the influence of the linker length; the increased length resulted in a decreased activity against AChE. On the other hand, the substitution of electron-donating and -withdrawing groups such as -CH_3_, -F, etc. plays a vital role in the hybrid performance. For example, methyl substitution on the phenyl ring improved the activity over that of the -F substitution. All the compounds showed good BACE-1 and Aβ inhibitions. The authors also employed molecular modeling studies to reveal the mode of interaction with target receptors. It is interesting to observe that the phenyl ring of piperazine interacts with the Phe20 residue via π–π stacking and carbonyl oxygen, and the amino group of piperazine forms hydrogen bonds with Lys16, which are significant for effective Aβ inhibition. The mixed type of interaction was identified by kinetic data. The neuroprotective activity and the BBB penetration abilities suggested that these hybrids deserve further in vivo studies [66]. To find out the substitution effect on the phenyl ring of the donepezil unit, a series of feruloyl–donepezil hybrids were developed by Viegas and coworkers (Figure 9B). The study showed that the substitution on benzylpiperidine decreases the inhibition of AChE, regardless of its size or electron-donating or -withdrawing abilities. The AChE inhibition of **22a**, IC_50_= 0.46 µM and **22b**, IC_50_= 16.74 µM and the selectivity of BuChE were very poor. The hydroxyl group of the feruloyl moiety confirmed the antioxidant activity (DPPH, EC_50_ = 49.41µM for **22a** and 46.66 µM for **22b**). Regiochemistry plays a major role in the antioxidant activity of the compounds [67]. Another series of donepezil–melatonin hybrids (**23a**–**e**) was synthesized by a fusion strategy, and their multitarget abilities were studied by Wang et al. (Figure 9C). These two pharmacophores were joined via ethyl acid carboxamide (**23a**,**b**), carboxamide (**23c**,**d**) and alkyl (2 carbon spacer) linkers. All the compounds exhibited micromolar to sub-micromolar inhibition of the ChEs (AChE, IC_50_ = 0.273 nM to 3.56 µM; BuChE, IC_50_ = 0.056 nM to 1.42 µM). Of note, the hybrids showed more potency toward BuChE compared to AChE. The substitutions of electronic properties and the position of the substitution on the phenyl ring influenced their inhibitory activity. While the two-position substituted methyl hybrids (**23a**, 1.54 µM for AChE and 1.42 µM for BuChE) showed better inhibition efficacy than the 2- or 3-Cl or -NO_2_ substitutions, which indicates that the electron-donating group and the position of the substitution have appeared more favorable in ChE inhibitions. However, increasing the linker length (**23e**) provided the strongest inhibition of ChEs (AChE and BuChE, IC_50_ = 0.27 nM and 0.056 nM, respectively). The anti-Aβ aggregation, antioxidant and BBB abilities showed potential multitarget molecules for the treatment of AD [68].

As the aforementioned hybrids significantly improved the AChE inhibitory activity in the presence of methyl (electron-donating group), a similar trend did not follow in all the cases. For example, Kong et al. developed a few donepezil–Trolox hybrids (**24a** and **24b**) with electron-donating -(CH_3_) and electron-withdrawing (-F) substitutions on the phenyl ring and introduced the alkyl linker between the Trolox and donepezil scaffolds (Figure 9D). The target hybrids were synthesized by the reaction of Trolox and the respective derivatives of donepezil in the presence of catalysts such as N-(3-(dimethylamino)propyl)-N′-ethyl carbodiimide hydrochloride (EDCI) and 1-hydroxy benzotriazole hydrate (HOBt). The results of the ChE inhibitory assay in vitro suggested that the rigid hybrid **24a** exhibited more selectivity in BuChE (IC_50_ = 5.49 µM) over AChE (IC_50_ = 19.5 µM). While the linker length was increased to the two-carbon spacer **24b**, the AChE inhibitory potency was much greater than **24a** (AChE, IC_50_ = 0.31 µM). In the case of MAO selectivity, the linker length did not influence the activity, and the most interesting factor was that the -F-substituted (electron-withdrawing group) hybrid showed better potential to inhibit AChE than the -CH_3_ (electron-donating group) group-substituted hybrids. Additionally, the antioxidant ability of **24b** was ABTS = 1.79 Tr and ORAC = 1.6 Tr, which was relatively close to the standard Trolox. The good neuroprotectivity, less cytotoxicity and BBB penetration abilities showed the value of the hybrids for AD [69]. Taken together, the inhibition potential changes based on the substituted functional groups and the position of a substitution on the MTDLs.

## 6. Effects of the Linkers and Substitutions on Tacrine-Derived MTDLs

Tacrine belongs to acridine compounds and is a potent noncompetitive reversible AChE inhibitor. Tacrine was the first medication licensed by the FDA for the treatment of AD. The compound was withdrawn from the market due to the severe hepatotoxic side effects. Numerous tacrine derivatives and hybrids showed excellent inhibitory activity for ChEs [70,71,72,73,74]. Bartolini et al. evaluated 26 tacrine–benzofuran hybrids against the key AD targets. The two chemical entities, tacrine and benzofuran, were joined via amido and aminoalkyl linkers (Figure 10A). All synthesized compounds showed better inhibitory activity against hAChE and hBuChE in a mixed-type mode. The hybrid with an amido linker having six carbon spacers displayed an improved potency of AChE inhibition (**25a**, IC_50_ = 424 nM) but had slightly decreased anti-AChE activity. The introduction of the methoxy group on the benzofuran moiety improved the potency (**25b**, IC_50_ = 24.4 nM). The varying results can be ascribed to the various positions of the benzofuran unit at the entrance to the AChE gorge, depending on the structure of the quinoline fragment. Interestingly, the introduction of an amino group in the spacer chain drastically increases the efficacy of the AChE inhibition (**25c**, IC_50_ = 0.86 nM), and it was 493 times more potent than tacrine. This could be explained by the fact that the amino group may involve π−cation interactions with the PAS, greatly enhancing the anti-AChE activity. On the other hand, with the introduction of another benzofuran moiety on the spacer, the activity decreased significantly (**25d**, IC_50_ = 260 nM) but showed better activity on BuChE (**25d**, IC_50_ = 0.48 nM). This indicates that the bulky group would not fit well into the AChE gorge, and as previously reported, the volume of the upper portion of the catalytic hBChE gorge is approximately 340 Å larger than the corresponding portion of the *h*AChE gorge, thus showing good selectivity in BuChE. Moreover, **25c** reduced the hAChE-induced amyloid fibrillization by 58% at a 100-mM concentration, which was comparable to bis(7)-tacrine (68%), and tacrine failed to inhibit AChE-induced Aβ_1__–40_ aggregation in similar experimental conditions. In contrast, **25d** showed better BACE-1 inhibition than other tested hybrids (IC_50_ = 0.19 µM) [75].

A novel series of tacrine–hydroxyphenyl benzimidazole hybrid molecules was synthesized and evaluated for their multitarget potency by Santos et al. The rationale for the design of the hybrids was as follows: the tacrine moiety was expected to provide AChE inhibition, and the hydroxyphenyl benzimidazole moiety should ensure Aβ inhibition and metal chelation; these two scaffolds were coupled with aliphatic linkers (Figure 10B). During AChE inhibition activity, the linker length, substitution on the linker and the tacrine moiety play a significant role. For instance, **26a** showed an IC_50_ value for AChE of 30.9 nM. It is worth noting that, by decreasing the linker length from 1 to 0 carbon spacers and -Cl atom substitution on tacrine, the activity was improved significantly (**26b**, AChE, IC_50_ = 6.3 nM). After the introduction of the -OH group on the linker, the activity (**26c**, IC_50_ = 16.8 nM) was lower compared to **26b,** and the activity further decreased while the -Cl substituted tacrine (**26d**, IC_50_ = 18.1 nM). These activity changes were explained by the molecular modeling data; the substituted chlorine atom fit well into the hydrophobic pocket of the AChE hybrids, where it was able to fit into the space formed by Trp432, Met436 and Ile439. The lengthening of the linker may keep away hydroxyphenyl benzimidazole from PAS. The installation of the -OH group on the linker may contribute to the improvement in enzyme inhibition via the H-bond with Asp72. In addition to that, the study confirmed that the hybrids were mixed-type inhibitors. The Aβ inhibition efficacy and the metal chelation abilities have added advantages to the hybrids [76].

Another series of electron-rich tacrine–multialkoxybenzene hybrids (**27a**–**d**) was synthesized, and their ChEs and Aβ inhibition abilities were evaluated (Figure 10C). It is interesting to note that the trimethoxy phenyl acetyl group linked to tacrine (**27b**, AChE, IC_50_ = 5.63 nM) showed better inhibitory activity than the dimethoxy-substituted (**27a**, AChE, IC_50_ = 65 nM) and hydroxy-substituted (**27c**, AChE, IC_50_ = 132.8 nM) hybrids, and a selectivity for AChE over BuChE was observed. The most potent inhibitor for AChE was **27b**, and its potency was 13 times stronger than the reference compound tacrine. However, a similar trend was not followed in Aβ inhibition (**27b**, Aβ_1–42_, IC_50_ = 51.8 nM and tacrine, IC_50_ = 12.21 nM). Despite that, the same electron-rich aromatic ring was afforded (**27b**), and the IC_50_ value for AChE of compound **27d** was decreased to 16.88 nM. A possible reason for this result could be that the additional vinyl group enlarged the conjugated regions in the same substituents of an aromatic ring, which meant that the electron density of the phenyl group was averaged and thus led to an electron-deficient effect. A Lineweaver–Burk plot and molecular modeling study proved that **27b** targeted both the CAS and PAS of ChEs [77].

## 7. Effects of the Linkers and Substitutions on Rivastigmine-Derived MTDLs

Rivastigmine is a carbamate derivative that is structurally related to physostigmine, which acts as a cholinergic agent for the treatment of mild-to-moderate dementia of the Alzheimer’s type. It is a reversible inhibitor that inhibits both cholinesterases (AChE and BuChE). Liu et al. synthesized a novel series of apigenin–rivastigmine hybrids and assayed their activity in vitro against AD targets (Figure 11A). In ChE inhibition, apigenin alone did not show any inhibitory activity. After the installation of a carbamate moiety into apigenin, the hybrids showed a significant ChE inhibition profile. This phenomenon shows the importance of the carbamate pharmacophore in ChE inhibition. Rat AChE inhibitory activity was changed based on the substitution of carbamate scaffolds; for example, arylamine substitution (**28a**, IC_50_ = 31.9 µM) showed a weak ratAChE inhibitory activity, and the activity was slightly improved with cyclic amine (**28b**, IC_50_ = 13.7 µM) substitution and showed better inhibition potency with aliphatic (**28c**, IC_50_ = 4.7 µM) substitution. The substituted aliphatic CH_2_ of **28c** interacted with key residue Trp86 via σ–π interactions, the carbonyl group interacted with the key Phe295 via intramolecular hydrogen bonding and the hydroxyl group (Apigenin) interacted with important residue Ser293 through intramolecular hydrogen bonding, which was identified by molecular modeling studies. Importantly, this carbamate moiety did not show any significant activity in antioxidant and Aβ inhibition. The parent apigenin scaffold plays a major role in antioxidant activity, particularly in the hydroxy group (**28c**, ORAC = 1 eq). Furthermore, the ideal neuroprotectivity, hepatoprotective activity and good BBB penetration abilities showed the potential of the hybrids against AD [78].

Other derivatives of rivastigmine-based MTDLs were synthesized by Deng and his coworkers (Figure 11B). The authors studied the biological activity of hybrids of ChE, antioxidant and neuroprotectivity. The results showed that all the compounds tend to inhibit ChEs and selectively prefer AChE over BuChE. The introduction of the N-ethyl- N-methylcarbamate side chain (**29a**) exhibited the strongest inhibition of BuChE with an IC_50_ of 6.2 µM. On the other hand, hybrids **29b** (IC_50_ = 0.34 mM) and **29c** (IC_50_ = 0.57 µM), containing the N,N-diethylcarbamate moiety, showed the most potent inhibition for AChE. It can be reasonably concluded that the N,N-diethylamine and N-ethyl-N-methylamine moieties fit better inside the gorge of the CAS in AChE. In general, when the N,N-diethylamine and N-ethyl-N-methylamine moieties were replaced by N,N-diisopropylamine and morpholine groups, the activity was decreased significantly. In addition to this, hybrid **29c** showed better antioxidant activity and caused less cytotoxicity. Overall, the aliphatic-substituted carbamates were the better choice than the aromatic or cyclic amine substitution [79].

## 8. Effects of the Linkers and Substitutions on Coumarin-Derived MTDLs

In recent years, coumarin derivatives have received a lot of attention because of their biological profiles related to AD [80]. Coumarins inhibit AChE by binding to the PAS of AChE. In addition to that, they act as potent MAO-B inhibitors by occupying the substrate cavity of MAO-B [81,82]. Rampa et al. synthesized a library of coumarin-based MTDLs for AD (Figure 12A). They introduced alkyl chains with various lengths and different terminal amino groups at the 6 and 7 positions of the previously described potent AChE inhibitor coumarin core (AP2243). Significantly, the diethylamino group (compound **30a**) installation at position 6 of the coumarin core led to improved potency on human AChE inhibition (IC_50_ = 11.7 nM) over the marketed drug donepezil (IC_50_ = 23.1 nM) and to a significant decrease in the self-aggregation of Aβ42 (~60%), whereas the replacement of the diethylamino group with a phenylpiperidine group (**30b**) decreased the hAChE inhibitory activity (IC_50_ = 4.1 µM) significantly and did not induce any activity on hBuChE and the self-aggregation of Aβ42. In addition, **30a** was able to block the neurotoxic effects induced by Aβ42 oligomers, thereby showing neuroprotective behavior, which made it a potential disease-modifying agent [83]. Kong et al. designed and synthesized a family of tacrine–coumarin hybrids with various lengths of linkers (Figure 12B). All the compounds exhibited sub-nanomolar inhibitory activities on ChE and micromolar activity on MAO. The most promising compound (**31a**) with three carbon unit lengths between tacrine and piperazine showed better inhibitory activity toward AChE (AChE, IC_50_ = 17.17 nM) than compound **31b** with a two-carbon unit linker (AChE, IC_50_ = 48.39 nM). However, this linker effect was not reflected in BuChE inhibition; the most potent compound was **31b** (IC_50_ = 16.12 nM). Most importantly, the bulky group substitution (phenyl) on coumarin reduced the ChE inhibition activity. On the other hand, the linker length did not show any significant profile on the MAO inhibition. The simultaneous substitution positions 3 and 4 at the coumarin moiety with the methyl group afforded the most potent MAO-B inhibitor, compound **31c** (MAO-B, IC_50_ = 0.24 µM), the inhibitory activity of which was about 10-fold more potent than that of compound **31a** (MAO-B, IC_50_ = 2.00 µM). The hybrids also exhibited low toxicity to SH-SY5Y cells, and in vitro studies confirmed their ability to penetrate the BBB [84].

## 9. Computational Strategies in MTDLs for AD Drug Design

In the continuous effort to discover potential MTDLs, computational approaches such as virtual screening, quantitative structure–activity relationships, molecular modeling approaches, machine learning, data mining and molecular simulations played a significant role in the identification of novel potential pharmacophores for various AD targets. We decided to describe some of the most representative recent examples. Mahboobeh et al. identified an MTDL (Figure 13A, **32**) through molecular docking, molecular dynamics (MD) simulation and binding-free energy calculations. The rational design of the MTDL adapts the combination of tacrine (AChE inhibitor) and modified dehydrozingerone (myeloid differentiation 2 protein inhibitor (MD2), which is necessary for TLR4 signaling), and the two chemical entities were linked with different lengths of linkers. The calculated binding energy of the designed MTDL (AChE, −13.0 and MD2, −9.4 kcal/mol) was much better than the standard drug tacrine (AChE, −8.1 kcal/mol and MD2- no data) in both enzymes. The results of the MD simulation confirmed that the designed ligand acts as a suitable inhibitor for AChE and MD2 proteins. Therefore, it can be applied as an effective antioxidant that can also target two other agents of AD appropriately [85]. Hui et al. designed two series of 26 tacrine–phenothiazine MTDLs for AChE and GSK-3β receptors. Based on the computational outputs, three compounds were selected for the synthesis. Initially, the fundamental physiochemical features were predicted for the designed 26 MTDLs by web-based PreADMET (v2.0), and the results showed the drug-likeness of the compounds. Further molecular docking trials were carried out by MolegroVirtual Docker (MVD) 2009 software. The outputs stated that compound **33** (Figure 13B) showed a better inhibitory profile against AChE and GSK-3β than the standard drug tacrine (MolDock Score of **33** toward GSK-3β was −148.821 kJ/mol; additionally, its MolDock Score toward AChE was −183.585 kJ/mol, tacrine AChE was −82.259 and GSK-3β was −71.497 kJ/mol). With the help of this assessment, the authors synthesized the compounds and performed biological experiments [86]. Kumar et al. synthesized a series of phenyl pyrazole derivatives to target AChE and MAO-B (Figure 13C). The structure–activity relationships indicated that chloro derivative **34** was a more effective AChE inhibitor as compared to the fluoro derivative, while the reverse trend was observed in MAO-B inhibitory activity. Though the potential binding orientations of the MTDLs to AChE and MAO-B were identified by molecular modeling, the experimental data did not stand in agreement with the computational data [87]. Very recently, Rehuman and co-workers designed halogenated coumarin-chalcone MTDL (**35**) and performed molecular docking studies against MAO-A/B and ChE (Figure 13D). The data showed that the coumarin ring involved π–π contacts with the hydrophobic region of FAD, Y407 and Y444 residues of MAO-B, and the styrene moiety interacts with the F208 residue. The hydrogen bond was identified between the hydroxyl group of the side chain of Y326 MAO-B and the carbonyl group of the chalcone bridge. In the case of AChE, the MTDL did not reach the CAS active site but interacted through F295 and W286 residues. On the other hand, π–π interactions observed in the catalytic W82 residue towards BChE are in agreement with the experimental data [88].

## 10. Conclusions and Future Perspective

This review provided the significance of linkers in the MTDL development for AD treatment, discussing the effects of the substitutions on the linkers and various positions of the MTDLs, rational design, synthetic conditions and biological evaluations of the most efficient MTDLs. In addition to that, the application of the computational approaches in MTDL development was discussed with examples. In fact, AD is one of many complex conditions that may require a multitarget drug strategy. Using similar approaches, numerous MTDLs have been synthesized for other multifactorial disease targets. For example, lapatinib is an anticancer agent that targets both human epidermal growth factor receptor 2 (HER2) and epidermal growth factor receptor (EGFR) tyrosine kinases [89,90]. Another example, Sotagliflozin, is an antidiabetic agent that inhibits both SGLT1 (sodium glucose co-transporter-1) and SGLT2 (sodium glucose co-transporter-2) [91,92,93]. In view of future developments, MTDLs will focus on a dual functional molecule targeting different systems of the body. One very interesting approach is Proteolysis Targeting Chimeras (PROTACs) that constitutes a novel and powerful method to degrade diverse proteins. PROTAC is a special kind of MTDL that combines a protein-binding ligand and an E3 ligase ligand connected into one molecule with a linker. PROTAC’s activity ultimately leads to ubiquitination, a naturally occurring process whereby proteins are not simply inhibited but are instead degraded into their constituent amino acids. The first oral PROTAC (ARV-110) targeting the androgen receptor progressed into a phase I clinical study in 2019 [94]. Recently, light-controlled PROTACs have been developed to degrade cancer-related lethal proteins [95]. Therefore, the future development and application of MTDLs will become expanded even more. It is justifiable to believe that the next generations of MTDLs will provide a pathway to treat complex diseases, including AD.

## Figures and Tables

**Figure 1 ijms-23-06085-f001:**
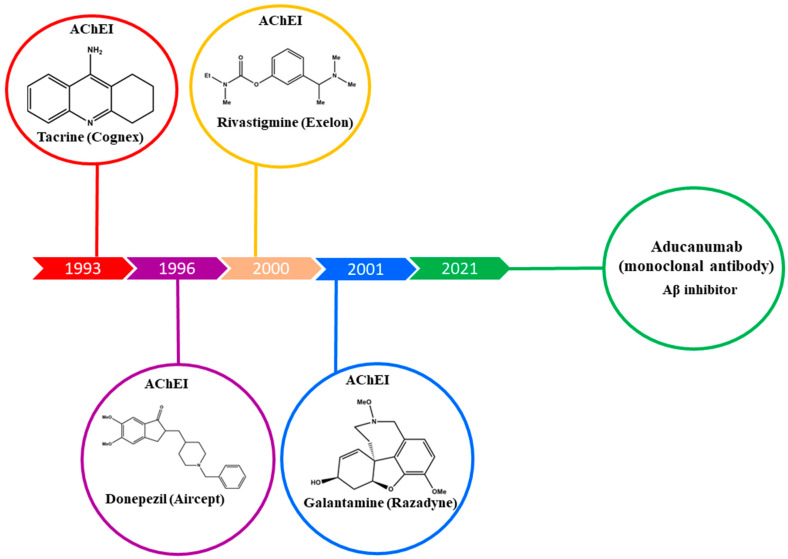
Current cholinesterase inhibitors and Aβ inhibitors are in clinical use (with trade names in parentheses), together with the years in which they received approval by the FDA.

**Figure 2 ijms-23-06085-f002:**
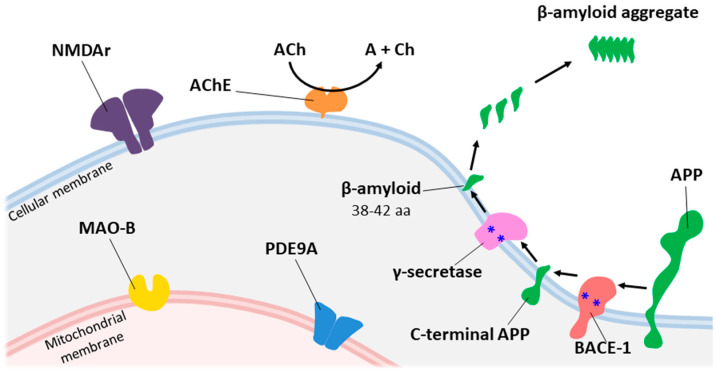
The major molecular targets in AD for MTDLs.

**Figure 3 ijms-23-06085-f003:**
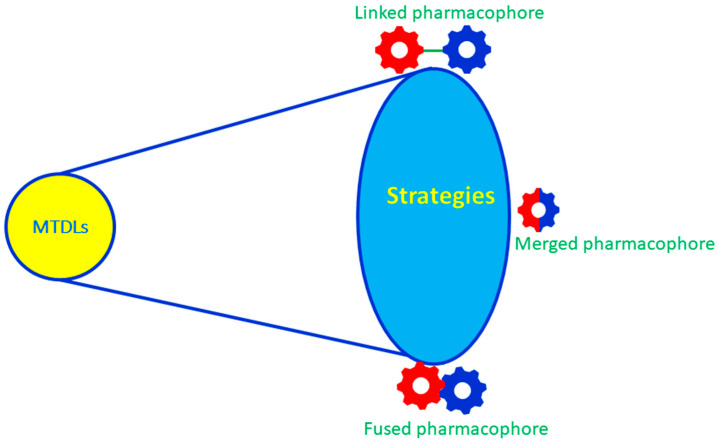
A schematic representation of the linked, fused and merged strategies in the MTDL designs.

**Figure 4 ijms-23-06085-f004:**
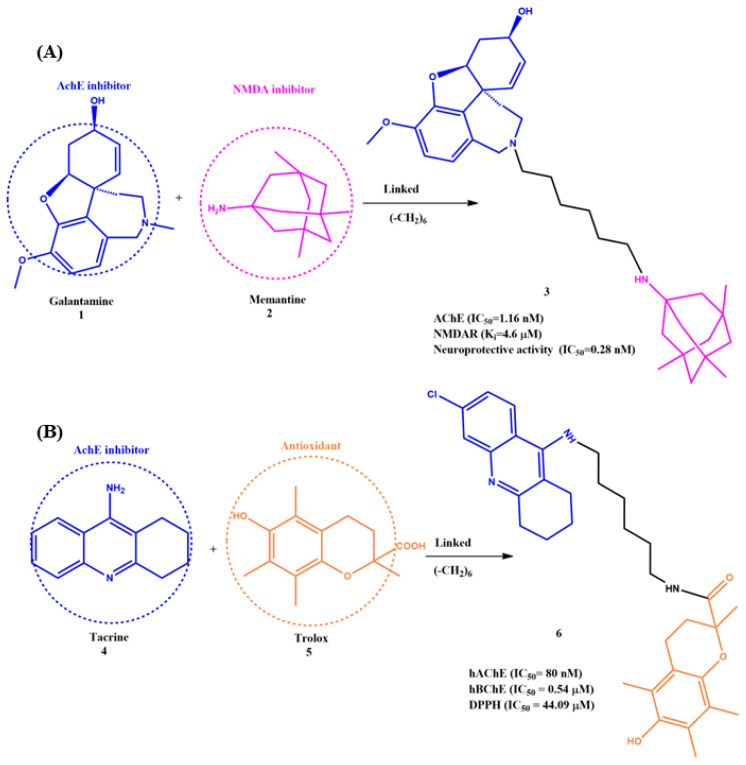
(**A**) and (**B**) Representative structure of linked pharmacophores.

**Figure 5 ijms-23-06085-f005:**
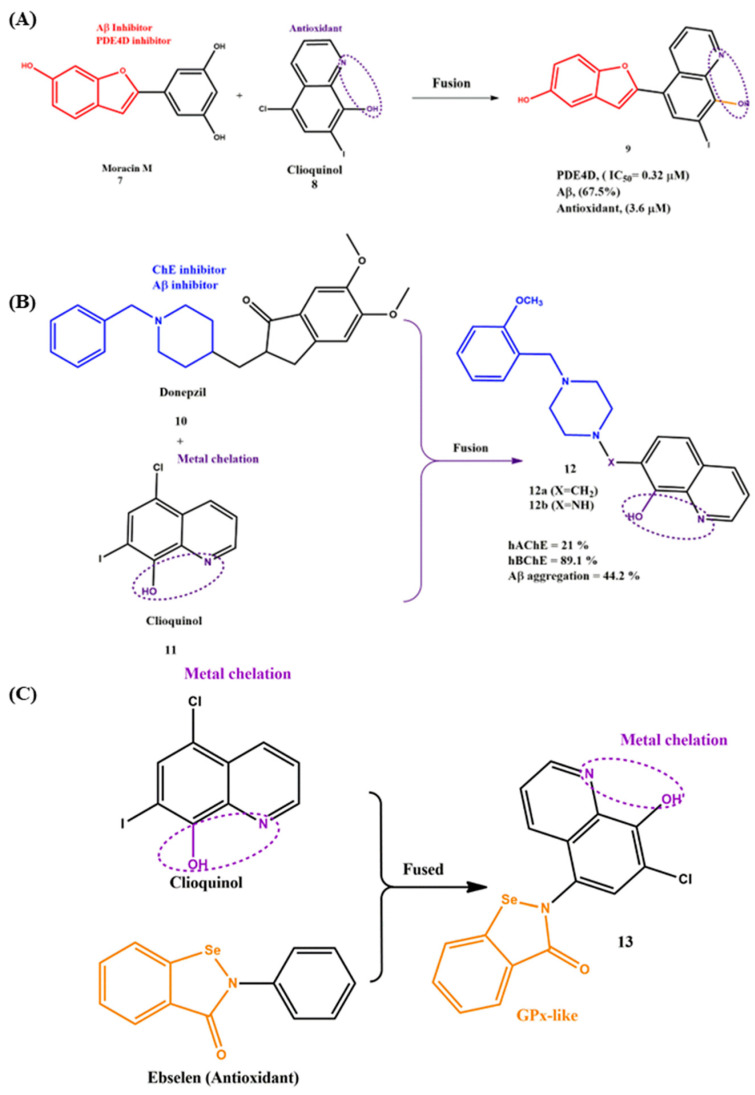
(**A**–**C**) Representative structure of fused pharmacophores.

**Figure 6 ijms-23-06085-f006:**
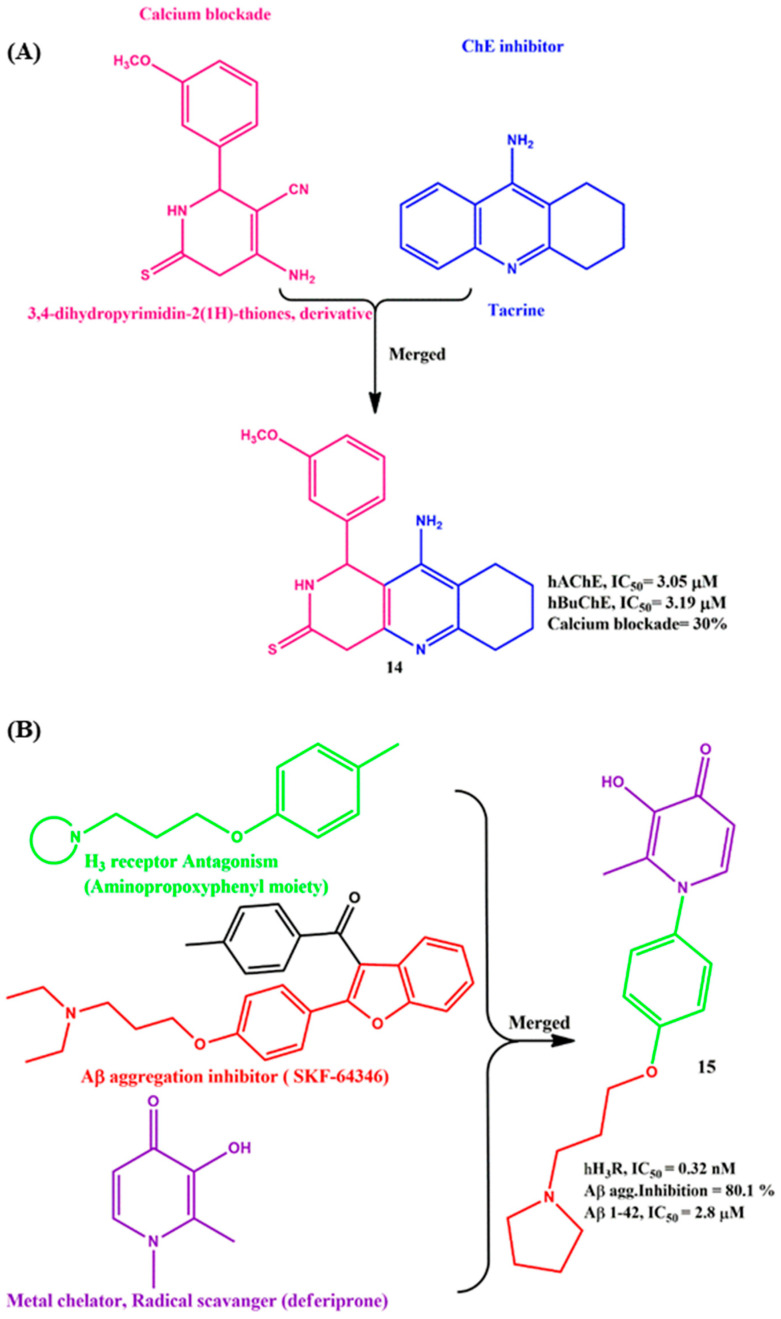
(**A**) and (**B**) Representative structure of merged pharmacophores.

**Figure 7 ijms-23-06085-f007:**
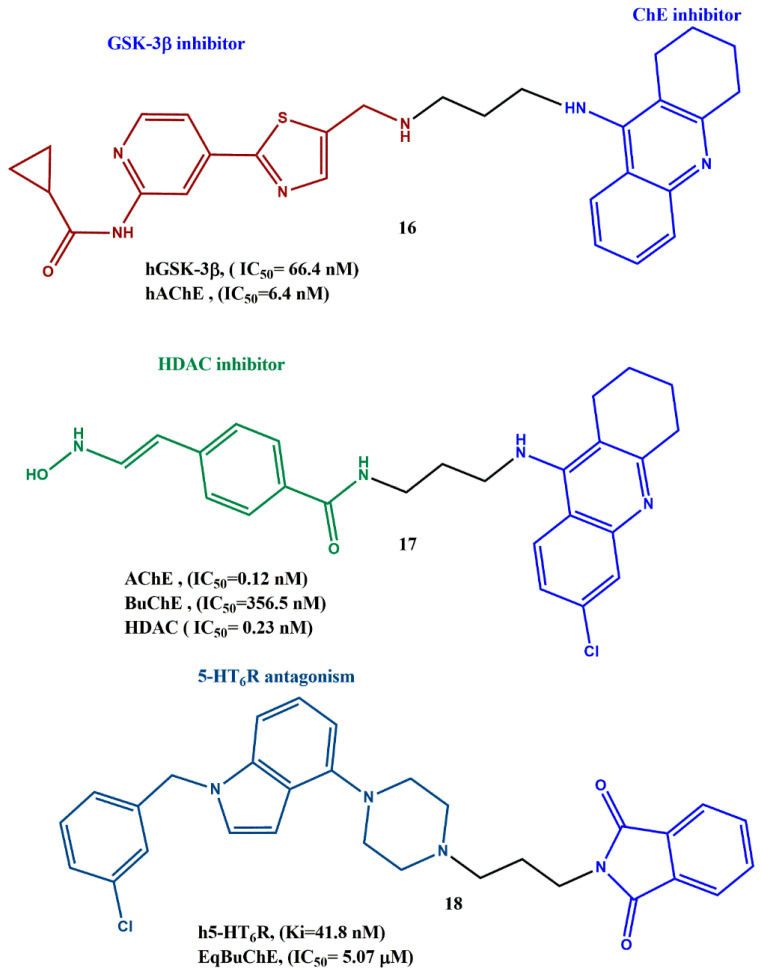
Examples of MTDLs for diverse AD targets.

**Figure 8 ijms-23-06085-f008:**
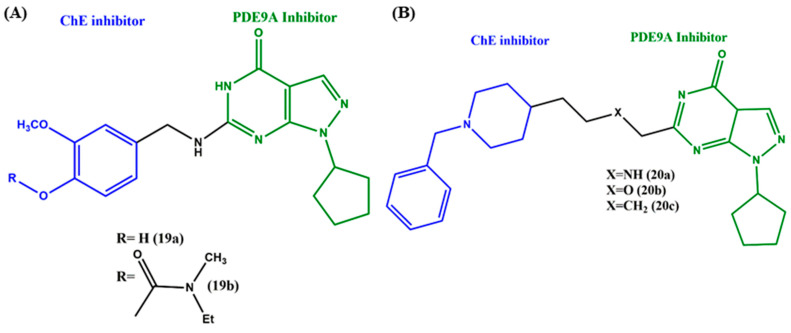
(**A**) and (**B**) Chemical structures of pyrazolopyrimidinone-derived MTDLs.

**Figure 9 ijms-23-06085-f009:**
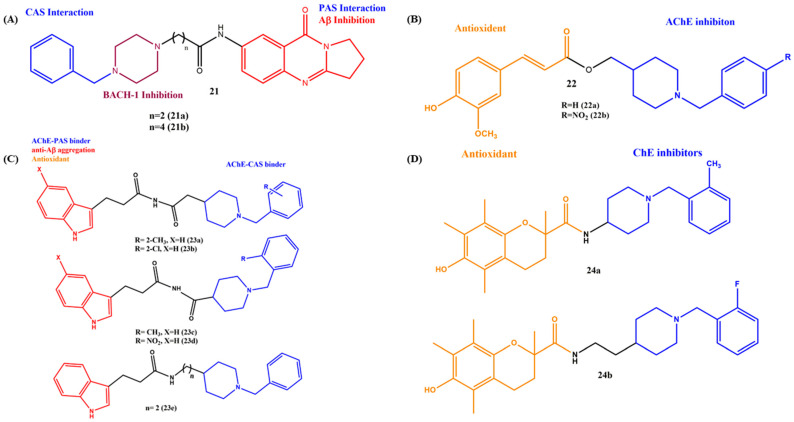
(**A**–**D**) Chemical structures of donepezil-derived MTDLs for AD.

**Figure 10 ijms-23-06085-f010:**
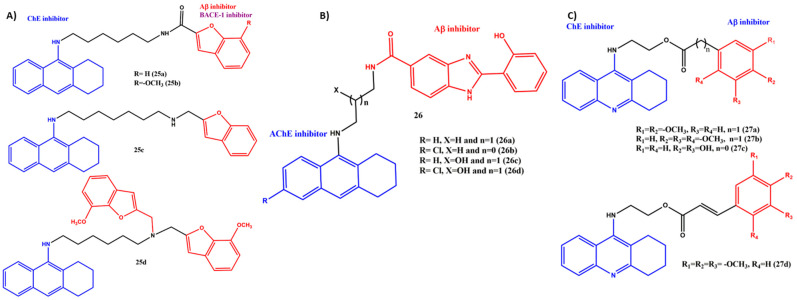
(**A**–**C**) Chemical structures of tacrine-derived MTDLs for AD.

**Figure 11 ijms-23-06085-f011:**
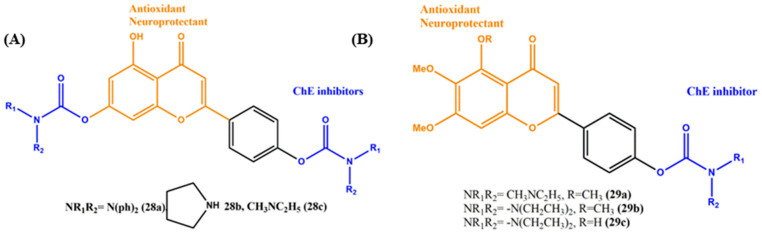
(**A**) and (**B**) Chemical structures of rivastigmine-derived MTDLs for AD.

**Figure 12 ijms-23-06085-f012:**
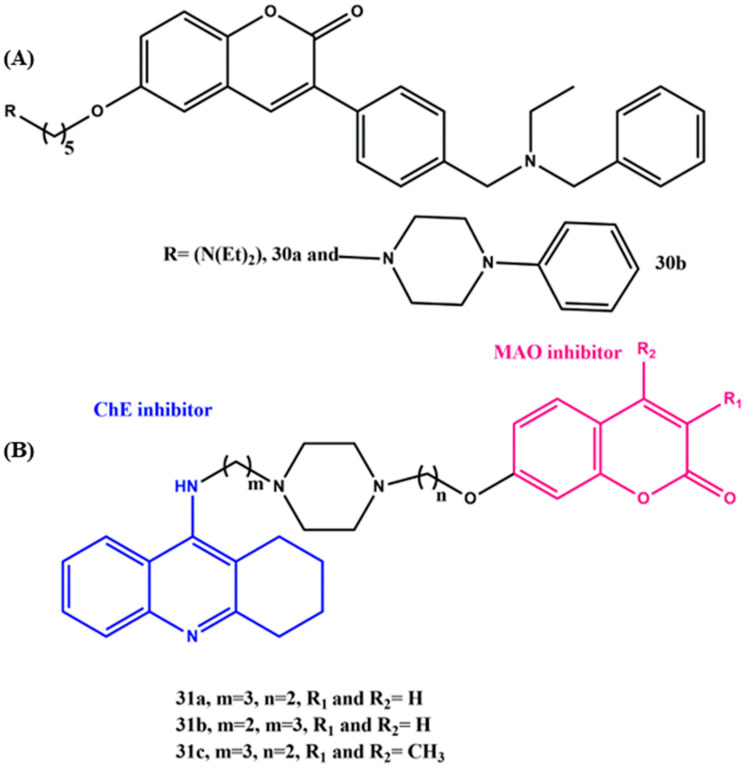
(**A**) and (**B**) Chemical structures of coumarin-derived MTDLs for AD.

**Figure 13 ijms-23-06085-f013:**
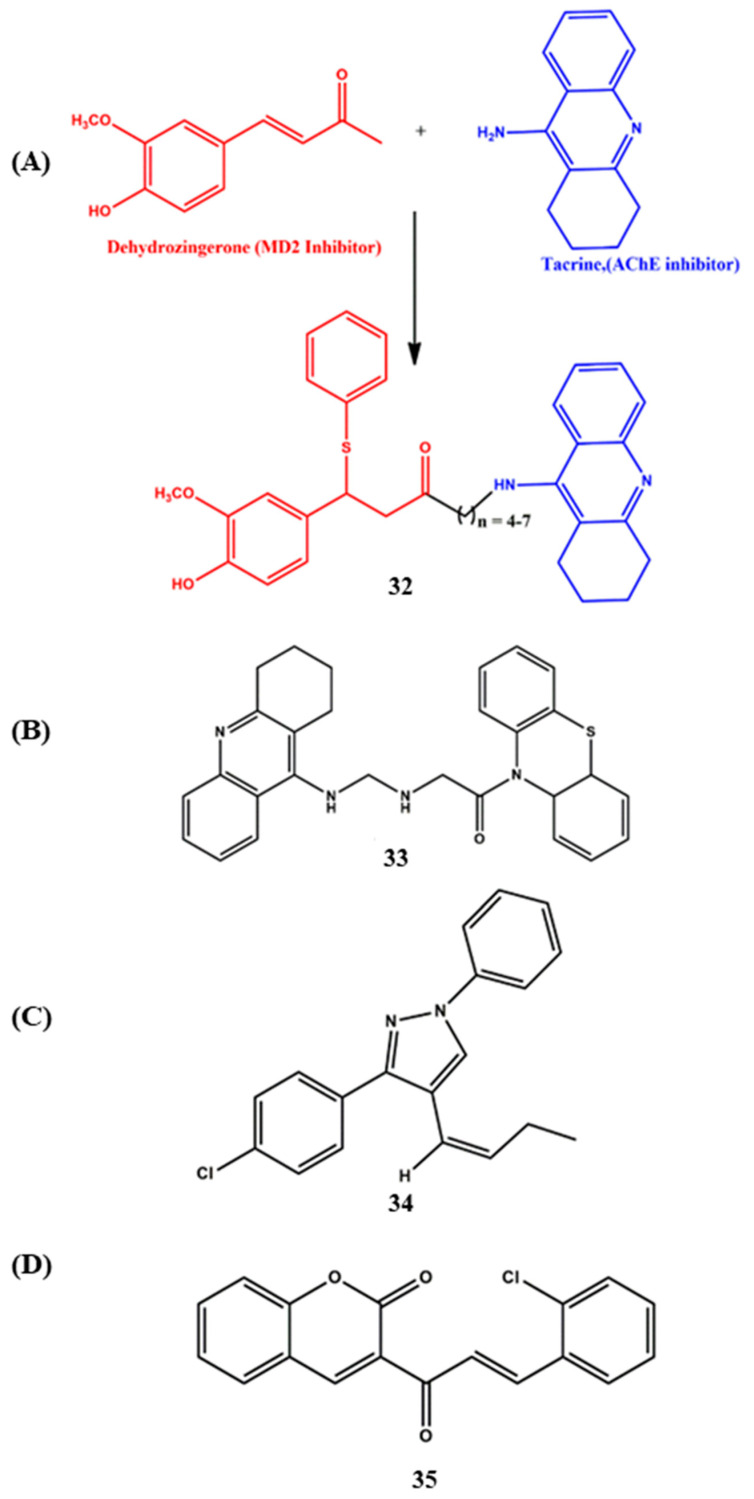
(**A**–**D**) MTDLs are designed using computational approaches.

## Data Availability

Not applicable.

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
