# Peer review of "Effects of Linkers and Substitutions on Multitarget Directed Ligands for Alzheimer’s Diseases: Emerging Paradigms and Strategies"

_ijms, 2022, doi:10.3390/ijms23116085_

Round 1
Reviewer 1 Report
A good paper, impressive in the insight offered into the mechanisms of pharmacological synthetic chemistry.
Introductory notions kind of long and general compared to the degree of detail present in the chemistry part.
The conclusions are practically non-existant. A summarization of the data brightly presented seems in order, no more than half a page.
Minor errors and confusions in the introductory part, as summarized in the annotated document.

Author Response
Comments and Suggestions for Authors
A good paper, impressive in the insight offered into the mechanisms of pharmacological synthetic chemistry.
Response: Thank you very much for the critical analysis of our manuscript. We highly appreciate your valuable comments, which indeed have considerably helped us to improve the quality of our manuscript. We have tried to address your concern and incorporated your suggestion in the revised manuscript.
Introductory notions kind of long and general compared to the degree of detail present in the chemistry part.
Response: As per your suggestion, the introduction part has been shorted in the revised manuscript.
The conclusions are practically non-existant. A summarization of the data brightly presented seems in order, no more than half a page.
Response: The conclusion part has been modified as per your suggestion. In particular we have indicated examples of other disease areas where multitarget directed ligands has been applied.
Minor errors and confusions in the introductory part, as summarized in the annotated document.
Response: All the errors and confusion in the introductory part have been corrected as per your suggestion.
Reviewer 2 Report
The present review article discusses the role of linkers and substitutions on multitarget-directed ligands in the context of Alzheimer’s disease (AD).
The review article is narrowly focused and could be of interest to the people working in the area of drug designing for AD. The authors have comprehensively covered the strategies utilized for designing linkers for multitarget-directed ligands for AD.
Below are my few comments:
In many places, authors have made some generalized statements, and have given only a single reference for those statements; authors are suggested to add multiple references for making a generalized statement.
Line 54-55: The increasing shreds of evidence indicate that 54 the multi-target approach may improve therapeutic efficacy and safety: Provide more references.
A cartoon diagram of the targets (e.g., choline acetyltransferase) showing the mode of drug binding should be shown for a better understanding to the reader.
All the figure legends should be described thoroughly (description of linkers, substitutions, etc.)
Though the article is focused on AD, the authors should also discuss other diseases where similar strategies have been applied to improve the drug's effectiveness.
Author Response
Comments and Suggestions for Authors
The present review article discusses the role of linkers and substitutions on multitarget-directed ligands in the context of Alzheimer’s disease (AD).
The review article is narrowly focused and could be of interest to the people working in the area of drug designing for AD. The authors have comprehensively covered the strategies utilized for designing linkers for multitarget-directed ligands for AD.
Response: We appreciate you for reviewing the manuscript and providing critical suggestions for our work. We are happy to hear that you find our work interesting.
Below are my few comments:
In many places, authors have made some generalized statements, and have given only a single reference for those statements; authors are suggested to add multiple references for making a generalized statement.
Line 54-55: The increasing shreds of evidence indicate that 54 the multi-target approach may improve therapeutic efficacy and safety: Provide more references.
Response: As per your suggestion, references have been provided in the appropriate places in the revised manuscript.
A cartoon diagram of the targets (e.g., choline acetyltransferase) showing the mode of drug binding should be shown for a better understanding to the reader.
Response: A cartoon diagram of major AD targets discussed in the text is included in the revised manuscript (new Fig.2.).
All the figure legends should be described thoroughly (description of linkers, substitutions, etc.)
Response: We agree with you that linkers and substitutions descriptions should be in the figure legends. All the reported compounds have various substitutions (Position and various functional groups) and linkers (length and various functional groups). In order to avoid confusion, we have described the substituted functional groups' on the figures and the details of the linkers were provided in the text.
Though the article is focused on AD, the authors should also discuss other diseases where similar strategies have been applied to improve the drug's effectiveness.
Response: As per your suggestions we have provided brief comment about other complex disease targets in the conclusion section of the revised manuscript.